# Quality Evaluation Based Simulation Selection (QEBSS) for analysis of conformational ensembles and dynamics of multidomain proteins

Amanda E. Sandelin [1,2], Ricky Nencini[3], Ekrem Yasar [4], Satoshi Fudo [1], Vassilis Stratoulias [2], Tommi Kajander [1] & O. H. Samuli Ollila [1,5] ✉

Multidomain proteins containing both folded and intrinsically disordered regions are crucial for biological processes, but characterizing their conformational ensembles and dynamics remains challenging. We introduce the Quality Evaluation Based Simulation Selection (QEBSS) protocol, which combines MD simulations with NMR-derived protein backbone $^{15}$N $T_1$ and $T_2$ spin relaxation times and hetNOE values to interpret conformational ensembles and dynamics of multidomain proteins. We demonstrate the practical advantage of QEBSS by characterizing four flexible multidomain proteins: calmodulin, EN2, MANF, and CDNF. These biologically important proteins have been difficult to study due to their flexible nature. Our findings reveal new insights into their conformational landscapes and dynamics, providing mechanistic understanding of their biological functions. QEBSS offers quantitative quality evaluation of simulations and a systematic approach for resolving conformational ensembles of multidomain proteins with heterogeneous dynamics. Given the importance of such proteins in biology, biotechnology, and materials science, QEBSS should benefit fields from drug design to novel materials development.

Most proteins consist of multiple domains whose mutual interactions and dynamics, such as hinge-bending and reorientations, often regulate their reactions and functions[1,2]. Experimental and computational characterization of folded proteins are well established[3] and tools to characterize disordered proteins are emerging[4–6], but these methods are typically not applicable for multidomain proteins containing both intrinsically disordered and folded regions. Tools enabling characterization of partially disordered proteins with significant flexibility will foster molecular level understanding of a wide range of systems with relevance in biology, biotechnology, and material science from cellular signaling[7] and therapeutic agents[8] to protein design for biomedical[9,10] and material applications[11].

Because nuclear magnetic resonance (NMR) spectroscopy is sensitive to fast timescale protein motions ranging from picoseconds to microseconds, it has been widely used to study motions of intrinsically disordered proteins (IDPs) and flexible multidomain proteins with intrinsically disordered regions (IDRs)[2,12–15]. Particularly longitudinal ($T_1$) and transverse ($T_2$) spin relaxation times and heteronuclear NOE (hetNOE) of protein backbone $^{15}$N are useful to probe molecular motions at nanosecond- and longer timescales[16]. In addition to dynamics, NMR spin relaxation times are sensitive also to conformational ensembles of partially disordered proteins[17]. However, interpretation of NMR spin relaxation data is often challenging and requires extensive modeling efforts, particularly for multidomain proteins with complex dynamics[17–19].

Molecular dynamics (MD) simulations have been demonstrated to be particularly useful for the interpretation of spin relaxation times from systems that exhibit too complex dynamics for other approaches, such as partially disordered proteins[17] and peptide-micelle systems[20]. Such approaches require careful validation against experiments because many force fields predict too condensed conformational ensembles for disordered protein regions[17,18,21] and wide conformational ensembles of disordered proteins may be difficult to fully sample by individual MD simulations. However, robust protocols to produce and select ensembles with most realistic conformations and dynamics of multidomain proteins from MD simulations have not been available.

[1]Institute of Biotechnology, University of Helsinki, Helsinki, Finland. [2]Division of Pharmacology and Pharmacotherapy, Faculty of Pharmacy, University of Helsinki, Helsinki, Finland. [3]Division of Pharmaceutical Biosciences, Faculty of Pharmacy, University of Helsinki, Helsinki, Finland. [4]Department of Biophysics, Faculty of Medicine, Erzincan Binali Yildirim University, Erzincan, Turkey. [5]VTT Technical Research Centre of Finland, Espoo, Finland. ✉e-mail: samuli.ollila@vtt.fi

Here we demonstrate that these challenges can be overcome by Quality Evaluation Based Simulation Selection (QEBSS), where conformational ensembles with most realistic dynamics are selected from diverse set of MD simulation data by comparing with experimental spin relaxation data. We demonstrate the practical advance of QEBSS by determining conformational ensembles and dynamic landscape of four multidomain proteins with heterogenous dynamics due to IDRs: calmodulin, Cerebral Dopamine Neurotrophic Factor (CDNF), Mesencephalic Astrocyte-Derived Neurotrophic Factor (MANF), and Engrailed 2 (EN2) (Fig. 1). Calmodulin, CDNF, and MANF consist of two folded or partially folded domains connected by a disordered and flexible linker. EN2 consist of one folded domain connected to a longer flexible, disordered linker (Fig. 1). In addition, the fully folded TonB C-terminal domain (TonBCTD)[22] has been analyzed for reference.

Calmodulin has two folded domains connected by a flexible linker and it samples different conformations in calcium dependent manner enabling binding to various targets with different structures[7]. For example, calmodulin regulates cardiac muscle contraction by binding to the RyR2 channel in different conformations[23]. MANF[24] and CDNF (Cerebral Dopamine Neurotrophic Factor)[25] are conserved homologous proteins with similar structures[26–28], both consisting of an N-terminal domain, a short linker, and a partially unfolded C-terminal domain with two alpha helices. The two domains of MANF and CDNF have been suggested to have distinct functions intra- and extracellulary[29], making the proteins bi-functional[26,30]. Both MANF and CDNF have shown cyto- and neuroprotective effects in Parkinson's disease models[25,31], and MANF in diabetes[29] and stroke[32]. CDNF has also been tested in Phase I clinical trials in Parkinson's disease patients[33]. EN2 is a functionally conserved protein needed for the formation of the midbrain/hindbrain border, an important organizer for neural development, and maintenance of mesencephalic dopaminergic neurons[34,35]. EN2 is characterized by five homology regions labeled EH1 to EH5, which exhibit significant conservation across species[36]. The region of interest for our modeling efforts spans residues 145 to 259, encompassing EH2 to EH4. Within this region, EH4 contains the homeodomain crucial for DNA binding, while EH2 and EH3 serve as Pbx-binding domains, playing a pivotal role in modulating EN2's DNA binding specificity and affinity[36,37].

Mechanisms of actions of these and other multidomain proteins are not fully understood partially due to limitations in available characterization methods for such proteins. Here, we present a method to explore conformational ensembles and interdomain dynamics in multidomain proteins with IDRs that can provide valuable insights into their poorly understood biological functions. Besides being a tool to characterize multidomain proteins with complex dynamics, QEBSS also provides quantitative quality evaluation of simulation trajectories with different force field parameters and initial configurations for different protein sequences. Together with machine learning and other modern computational tools, such information is expected to lead to major advances in modeling and understanding of disordered biomolecular complexes, as already demonstrated for lipids[38,39].

## Results

### Production of multidomain protein conformational ensembles with realistic dynamics

To determine conformational ensembles and dynamics for selected multidomain proteins, namely *Xenopus laevis* calmodulin, human CDNF, mouse MANF, and chicken EN2, as well as the folded C-terminal domain of *Helicobacter pylori* TonB (TonBCTD) as a control, we first ran 1 microsecond MD simulations with leap-frog algorithm from five different starting structures using five force fields (a99SB-ILDN[40], DESamber[41], a99SB-disp[21], aff03ws[42], a99SBws[42]), producing a total of 25 simulations for each protein. The produced trajectories were then evaluated against experimental spin relaxation data. For full details about initial conformations and simulation details, see steps 1 and 2 in QEBSS protocol described in the "Methods" section.

The selected force field parameters are specifically designed for disordered proteins and are expected to provide the best descriptions for IDRs

among the available parameters[17,21] with the exception of a99SB-ILDN, which is a standard force field for folded proteins. Conformational ensembles from these simulations are illustrated in terms of overlayed snapshots in Fig. 1, radius of gyration distributions in Supplementary Figs. S1–S5, and backbone orientational correlation maps in Supplementary Figs. S6–S10.

As expected, limited conformational flexibility and minor differences between force fields and replicas are observed in simulations of the folded TonBCTD (Supplementary Figs. S1, S6, and S18). On the other hand, the studied multidomain proteins with disordered regions exhibit wider conformational sampling and larger variations between force fields and replicas (Supplementary Fig. S18). In line with previous studies[17,21], a99SB-ILDN force field optimized for folded proteins predicts more compact ensembles for CDNF, MANF, and EN2 than force fields optimized for disordered proteins (Fig. 1), which is also visible as stronger interdomain correlations in Figs. S8–S10. However, this trend is not observed for calmodulin. Radii of gyrations averaged over simulation replicas do not show major differences between force fields for multidomain proteins (Fig. 1), but overlayed snapshots (Fig. 1), radius of gyration distributions (Supplementary Figs. S1–S5), backbone orientational correlation maps (Supplementary Figs. S6–S10), and principal component analysis (PCA, Supplementary Fig. S18) reveal differences between results from different simulation parameters and starting structures.

To evaluate the quality of conformational ensembles and dynamics in simulations, we calculated backbone $^{15}N$ spin relaxation times, $T_1$ and $T_2$, and hetNOE values separately from each trajectory (QEBSS step 3 described in "Methods" section). Spin relaxation times from different force fields, calculated from correlation functions averaged over different replicas, are compared with experimental spin relaxation data[13,28,43–45] in Fig. 2, and individual replicas in Supplementary Figs. S11–S15. Significant differences in qualities are observed between different force fields and replicas. In some cases specific force fields seem to overcome others in quality, such as a99SB-ILDN for TonBCTD and calmodulin, but often average qualities of force fields are difficult to rank despite their differences, which is the case for example for MANF. On the other hand, differences between individual replicas are often larger than differences between averages over force fields. This indicates that insufficient sampling of conformations during simulations affects the results, emphasizing the importance of launching simulations from different starting configurations.

In conclusion, currently available force fields are able to produce dynamical ensembles that reproduce experimental spin relaxation times at least for some multidomain proteins. These ensembles can be used to provide interpretation for MD that correspond the experimental spin relaxation data. However, none of the force fields generally outperforms others, and best parameters depend on the protein under interest (Fig. 2). It is therefore difficult to know beforehand which parameters are most realistic for a given protein. On the other hand, results also depend on initial configurations.

### Selection of most realistic conformational ensembles of multidomain proteins

To find the most realistic ensembles among generated MD simulations (QEBSS step 4 described in "Methods" section), we first calculated the root-mean-square deviation (RMSD) averaged over residues between $T_1$, $T_2$ spin relaxation times and hetNOE values from all simulations and experimental data (Supplementary Tables S1–S5). Simulation trajectories for which RMSDs averaged over residues deviated less than 50% from the best simulation for all spin relaxation times ($T_1$, $T_2$, and hetNOE) were then selected for further analysis. It is essential to select simulations that simultaneously reproduce all experimental spin relaxation times because they are sensitive to different timescales.

Different RMSDs for all replicas are listed in Supplementary Tables S1–S5, where simulations selected based on our criteria are bolded (a total of nine simulations of TonBCTD, two of calmodulin, 13 of CDNF,

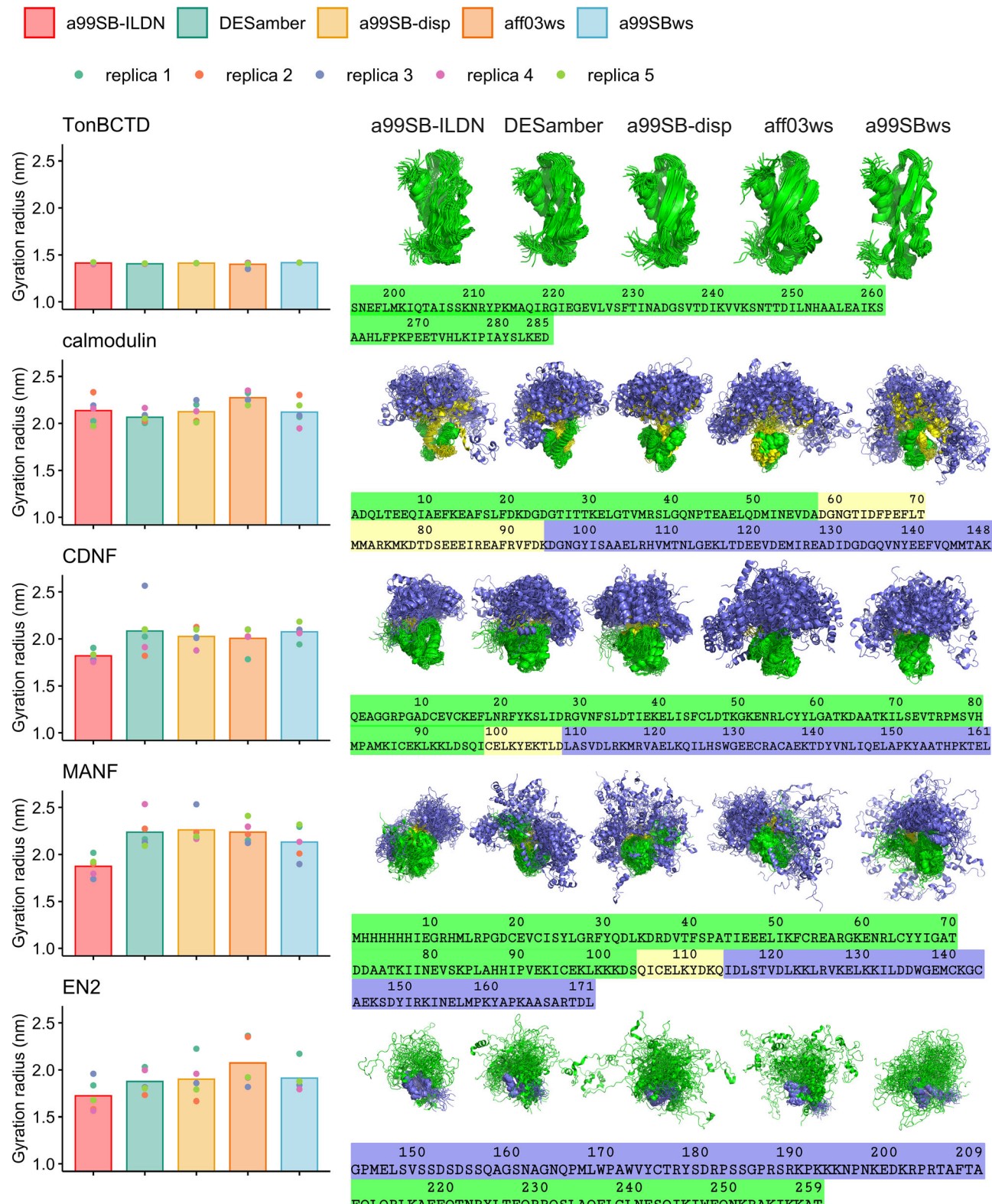

**Fig. 1 | General characterization of simulations.** Average radius of gyration (nm) for each force field calculated from five replicas (*left*). Representative snapshots showing 50 overlaid structures per force field, with 10 equally spaced structures taken from each of the five replicas. For structural alignment, the entire protein was used for TonBCTD[22], while the N-terminal folded domains were used for calmodulin[60], CDNF[28], and MANF[44], and the C-terminal domain for EN2[45]. Domain organization, sequence, and residue numbering of each protein are shown to illustrate the different domains of the multidomain proteins (*right*).

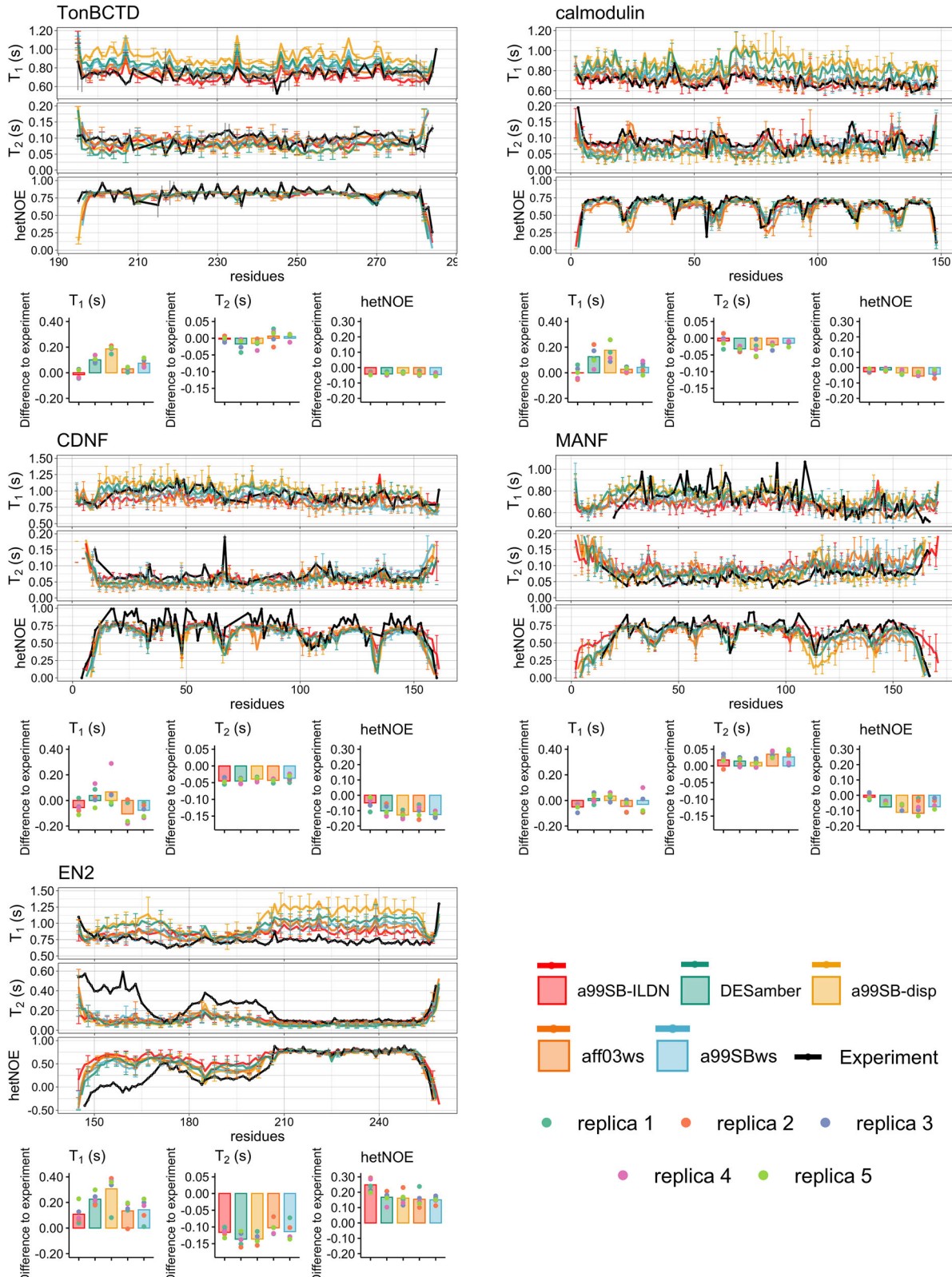

**Fig. 2 | Comparison of spin relaxation times predicted by different force fields with experiments.** Backbone $^{15}$N spin relaxation times, $T_1$ and $T_2$, and hetNOE values calculated from averages over replicas from MD simulations with different force fields compared to experimental spin relaxation data for TonBCTD[43], calmodulin[13], CDNF[28], MANF[44], and EN2[45]. Average difference over all residues between calculated $^{15}$N spin relaxation times, $T_1$ and $T_2$, and hetNOE values from simulations and experimental values. Experimental errors are displayed with gray error bars if reported.

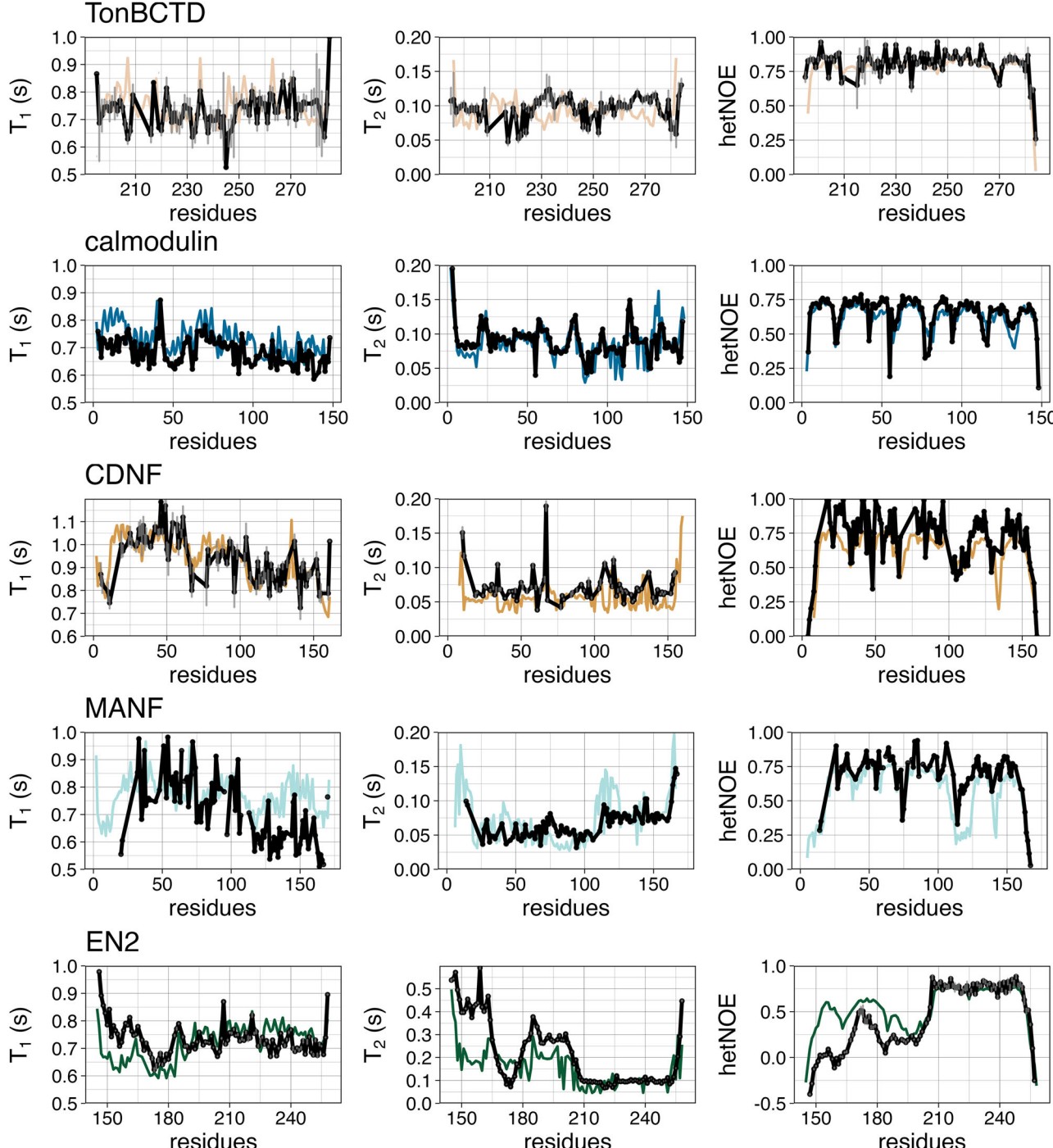

**Fig. 3 | Comparison of spin relaxation times between QEBSS ensemble and experiments.** [15]N spin relaxation times, $T_1$ and $T_2$, and hetNOE values calculated from averaged correlation functions from the total QEBSS ensemble resulting from the QEBSS protocol for different proteins. Experimental errors are displayed with gray error bars if reported.

three of MANF, and two of EN2). Spin relaxation times of the QEBSS ensemble were calculated by averaging correlation functions of the chosen simulations and are compared with experiments in Fig. 3 and the RMSDs of the QEBSS ensembles are shown in Supplementary Tables S1–S5. Because the total QEBSS ensembles composed by combining the selected simulations exhibit better or similar quality against experiments when compared with the best individual simulations, we consider the total QEBSS ensembles as the best possible interpretations for conformational dynamics observed in the experiment. Even if an individual simulation would have a slightly better ranking, we consider the total QEBSS ensemble as a better choice because it

always contains more conformations, thereby covering a larger volume in the phase space, than an individual simulation.

Selected simulations originate from more than one force field for all proteins except MANF for which all three selected trajectories were simulated with a99SB-disp parameters (Supplementary Tables S1–S5). On the other hand, all replicas of a particular force field were not selected for any of the studied proteins. This emphasizes the importance of using different parameters and starting configurations to generate sufficiently wide conformational distributions with the currently available resources and parameters.

Notably, simulations with the best scores for individual spin relaxation times may not be selected for further analyses based on our criteria. For example, neither of the selected calmodulin simulations had the lowest RMSD value for $T_1$, $T_2$, or hetNOE, but the two selected simulations were the only simulations with all the values simultaneously close to the experimental data. In this study, we set the QEBSS threshold at 50% to ensure that each protein system had at least one simulation included in the QEBSS ensemble. Future studies may benefit from optimizing this threshold based on the quality of simulation and experimental agreement.

Overall agreement of QEBSS ensembles and experiments is very good in Fig. 3, with the exceptions of EN2 simulations, where $T_2$ values are slightly underestimated and hetNOE values overestimates for the flexible linker, and of MANF, for which some deviations are observed for residues 100–171. These differences highlight the importance of developing more accurate force field parameters that would create more realistic ensembles for the QEBSS protocol. In the case of μs to ms timescale conformational exchange, $T_2$ values may be affected by dynamic phenomena that are not captured by the analysis used in this work[46], leading to discrepancies in $T_2$ values with respect to other spin relaxation times. Nevertheless, such situation is not observed for systems analyzed here as all spin relaxation time values from QEBSS ensembles are in good agreement with experiments in Fig. 3.

Notably, QEBSS ensembles determined in this work are in good agreement also with experimental small angle x-ray scattering (SAXS) data (supplementary Fig. S16) and spin relaxation data from magnetic fields that are not used in QEBSS (supplementary Fig. S17).

Because characterization and mutual comparisons of conformational ensembles is complicated, differences between ensembles accepted or rejected by QEBSS are not always straightforward to explain. Nevertheless, distinctive features in radius of gyration distributions are observed in all proteins except CDNF, where multiple types of distributions appear across the selected simulations (Supplementary Figs. S1–S5). To quantify differences between simulation ensembles, we performed PCA for all simulations (Supplementary Fig. S18). For EN2 and calmodulin, their selected ensembles are closely positioned in PCA space, indicating structural similarity (Supplementary Fig. S18). Substantial similarities for the second principal component of QEBSS ensembles of MANF are observed, while CDNF ensembles are more diverse, as observed also for radius of gyration distributions. As expected, variance in PCA for folded TonBCTD is substantially smaller than for proteins with flexible IDRs.

Furthermore, the selected QEBSS ensembles differ significantly from the ensemble used to select the starting structures (Supplementary Fig. S19), supporting the conclusion that the QEBSS protocol provides new insights into protein ensembles by providing interpretation for experimental spin relaxation data.

## Analysis of conformational ensembles and dynamic landscapes of multidomain proteins

Conformational ensembles resulting from the QEBSS procedure reproduce spin relaxation times in good agreement with experiments in Fig. 3. Therefore, we use these results to interpret ensembles and dynamic landscapes of studied multidomain proteins and compare them with each other (QEBSS step 5 in the "Methods" section). Radius of gyration distributions, overlayed snapshots, and dynamic landscapes, in terms of weights of different timescales in Eq. (1), resulting from QEBSS ensembles are shown in Fig. 4. Additionally, the average orientational correlation and minimum distance between each residue and effective correlation times from QEBSS ensembles are shown in Fig. 5.

**TonBCTD**. TonBCTD exhibits typical behaviour for a folded protein composed of 92 residues with a narrow and symmetric radius of gyration distribution, clear dominant timescales around 5 ns with weights over 0.75 for most residues corresponding to overall rotation of the protein, and very fast timescales of <10 ps with weights of 0.10–0.15[47]. The dynamic landscape of TonBCTD is distinct compared to that of calmodulin, CDNF, and MANF, which are composed of two folded domains

that are attached by a linker. EN2 is a partially disordered protein where the dynamics of the folded region is similar to globular proteins such as TonBCTD, while the disordered region exhibits heterogeneous dynamic landscape[17].

**Calmodulin**. Results in Figs. 4 and 5 suggest that the N-terminal and C-terminal domains of calmodulin do not directly interact, their movement with respect to each other is not fully flexible and that the latter exhibits more disordered characteristics. These results are supported by the orientational correlation and minimum distance analysis in Fig. 5, indicating that the two folded domains of calmodulin are partially correlated, but that there are no clear contacts between the domains. Moreover, the first 75 residues in the N-terminal domain of calmodulin exhibit dominating rotational timescales between 6 and 12 ns with total weights of around 0.7–0.8 and faster timescales below 2 ns with weights from 0.2 up to 0.4 (Fig. 4 and Supplementary Fig. S20). In the C-terminal domain of calmodulin, timescales between 2 and 6 ns with varying weights are also observed for many residues, indicating more heterogeneous dynamics than in the N-terminal part. Such heterogeneous dynamics is characteristic for disordered protein regions[17]. These results are in line with previous interpretations of spin relaxation times based on the fitting of simple rotational models to experimental data and interpretations of higher hydrogen exchange rates in the C-terminal domain[12,13]. Nevertheless, our atomistic model from QEBSS provides higher resolution information than common rotational timescales for all residues in previous studies. Good agreement with the experimental radius of gyration[48] and SAXS data[49] (supplementary Fig. S16) gives further support for the validity of the QEBSS ensemble.

**MANF and CDNF**. MANF and CDNF have been hypothesized to have similar mechanism of action due to structural similarity[8]. However, QEBSS results in Fig. 4 reveal some differences between the proteins' conformational dynamics. CDNF exhibits slower overall dynamics (dominant timescales between 10 and 16 ns with weights above 0.75 in the N-terminal domain and 6–12 ns with weights 0.5–0.75 in the C-terminal domain) than MANF (dominant timescales between 6 and 12 ns with weights of 0.6–0.8 for all residues). On the other hand, the C-terminal domain and linker region in MANF have higher weights for fast timescales below 2 ns compared to the N-terminal domain, while such difference is not observed for CDNF (Supplementary Figs. S20–S21). Altogether, our results in Fig. 4 and Fig. 5 indicate more correlated rotations and interactions between two domains in CDNF than in MANF where domains interact less and rotate more freely with respect to each other. The correlated rotations of two domains, forming a rigid-body-like object, explain the slower overall dynamics of CDNF than MANF where domains rotate more independently. The observed differences in conformational ensembles and dynamics may imply distinct mechanisms of action for these proteins despite their structural similarity. Although human MANF and human CDNF share 55.93% of their amino acid sequence, only human MANF can rescue the lethality of MANF knockout in Drosophila melanogaster[50]. Additionally, CDNF stands out as the only one expressed in muscles[51,52], a tissue microenvironment characterized by high ATP levels and significant variations in calcium and oxygen. However, it is important to note that, due to the availability of experimental data, we compare mouse MANF with a His-tag to human CDNF in the absence of His-tag. Nevertheless, minor differences in the sequence or His-tag with 10 residues are not expected to have a major effect on the results.

**EN2**. EN2 exhibits a typical dynamic landscape for a partially disordered protein where the folded C-terminal domain resembles rigid body rotations, while the unfolded N-terminal domain has more heterogeneous dynamics[17] (Fig. 4). The N-terminal domain displays faster dynamics with <2 ns and 2–6 ns timescales having weights around 0.3–0.6 compared to the folded C-terminal domain with high weights

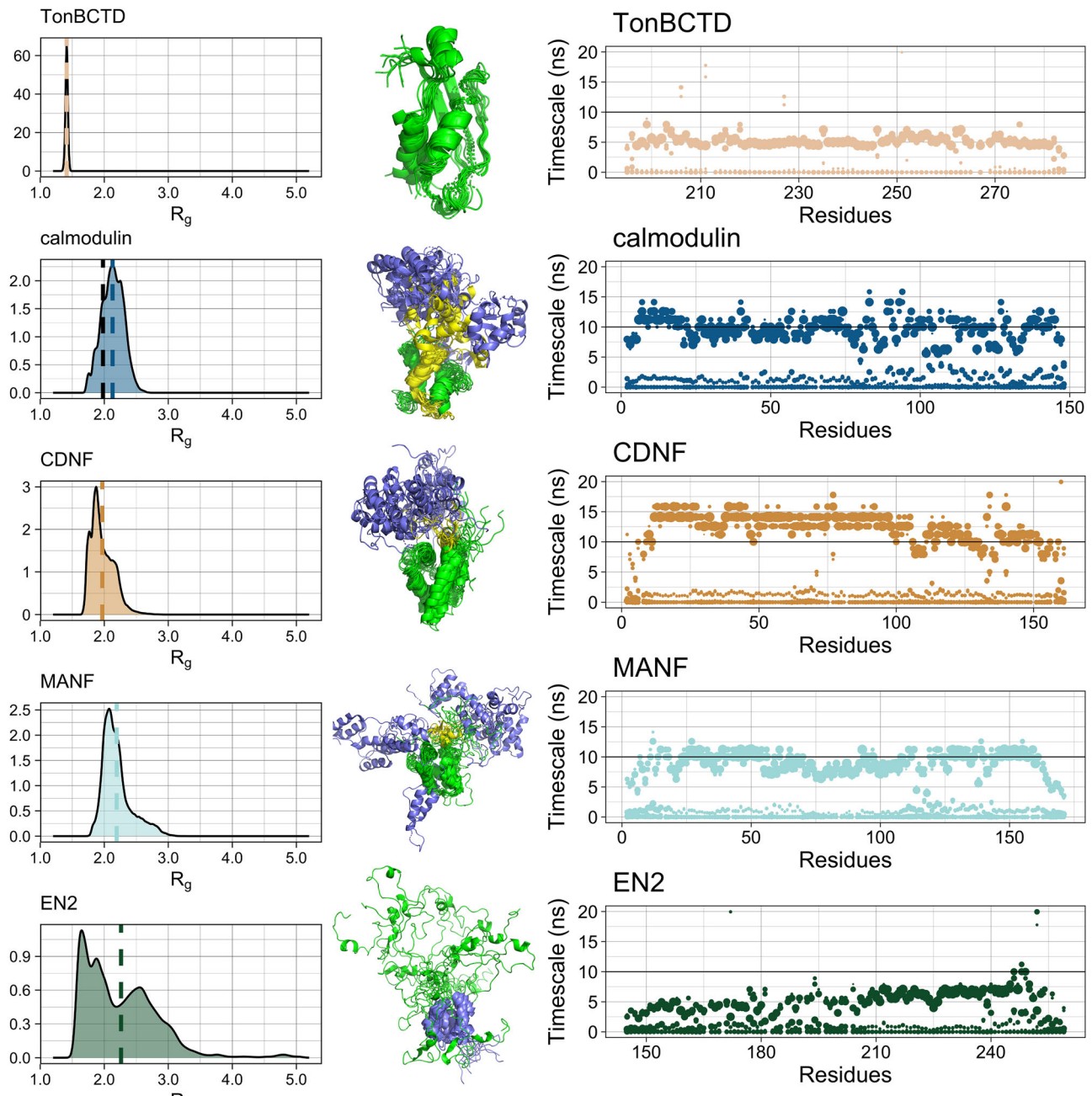

**Fig. 4 | Characterization of overall shapes and timescale distributions of QEBSS ensembles.** Radius of gyration ($R_g$ in nm) distributions (*left*), snapshots displaying 10 overlayed structures (*middle*), and timescales (*right*) from QEBSS ensembles for different proteins. Dashed vertical lines show the average $R_g$ and the experimental value for calmodulin[48] is marked with a black dashed line. For the snapshots, structural alignment of the entire protein was used for TonBCTD[22], while the N-terminal folded domains were used for calmodulin[60], CDNF[28], and MANF[44], and the C-terminal domain for EN2[45]. The point sizes represent the weight of each timescale in the rotational relaxation process.

(0.8) for slower 6–8 ns timescale and lower weights (0.2) for fast <2 ns timescales (Fig. 4 and Supplementary Figs. S20–S21). In line with previous study[17], our results suggest that folded and unfolded domains rotate independently from each other. However, underestimated $T_2$ values and overestimated hetNOE values for the disordered N-terminal domain (Fig. 3) suggest that the domain is still probably too ordered also in the QEBSS ensemble. On the other hand, the selected simulations for EN2 have a wider radius of gyration distributions than other simulations (Fig. S5), indicating that MD simulations tend to predict too compact conformational ensembles for partially disordered proteins in line with previous studies[17,21]. Also, the QEBSS ensemble of EN2 has worse agreement with experiments than other proteins. In a previous study,

simulation results were optimized by mutating amino acids that were known to have poor performance in the used aff03ws force field[53], which resulted in a slightly better agreement for EN2 disordered region[17]. This indicates that with the current force fields, it may be more challenging to find good QEBSS ensembles for partially or fully disordered proteins than proteins with multiple folded domains attached by flexible linkers. Nevertheless, our results here and in the previous study demonstrate that sufficient accuracy for the interpretation of experiments can be reached[17].

## Discussion
Based on our results, we propose here the QEBSS protocol to determine conformational ensembles and dynamics of multidomain proteins with

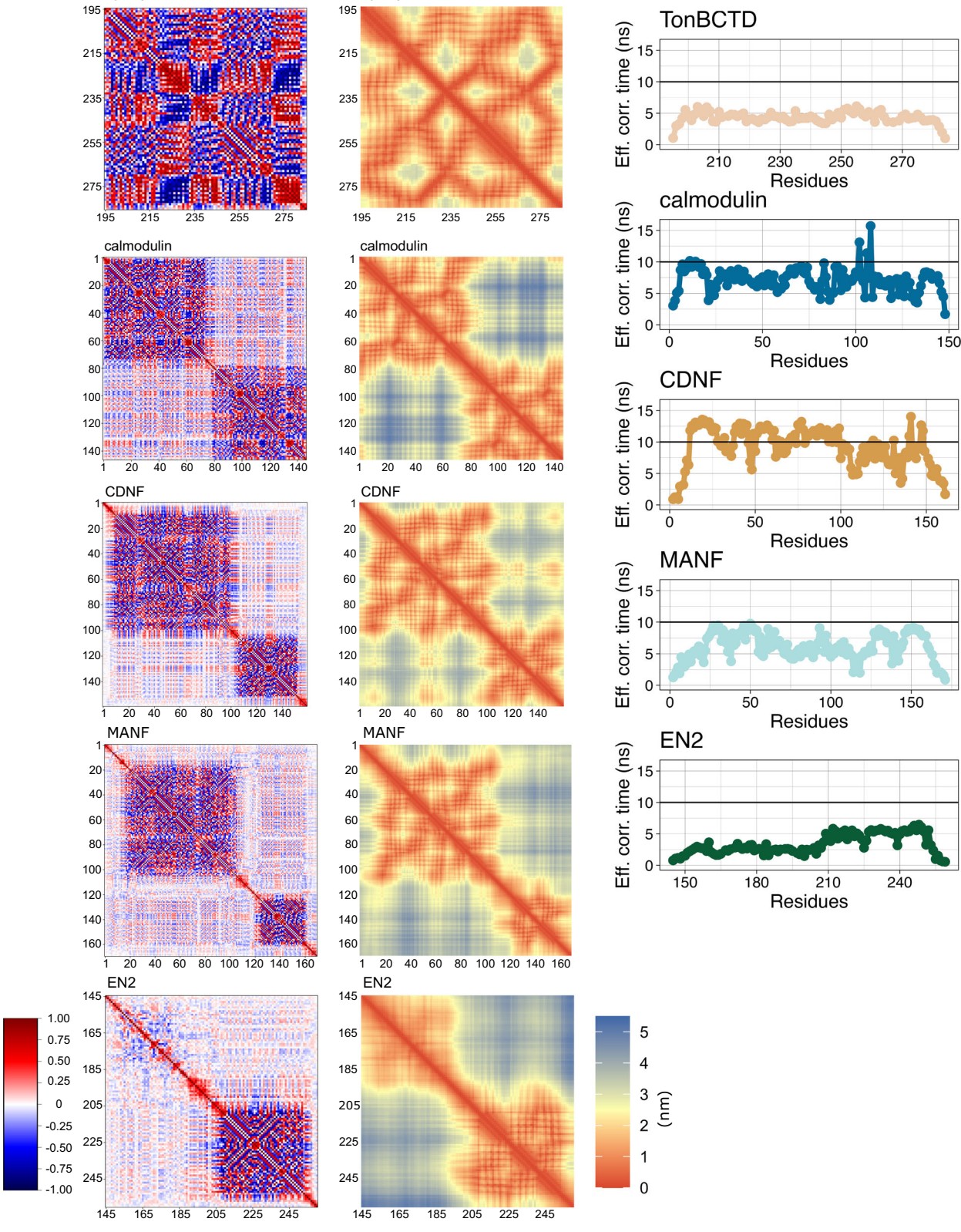

**Fig. 5 | Characterization of backbone correlations, contacts, and overall dynamics of QEBSS ensembles.** Protein backbone orientation correlation maps for vectors between Cᵅ carbons of consecutive residues (*left*), average minimum distance between residues maps (*middle*), and effective correlation times calculated from QEBSS ensemble for different proteins (*right*).

IDRs. Steps in this approach are described in detail in "Methods" section and can be summarized as: (1) choosing a diverse set of starting structures, (2) generating MD simulation trajectories with different force fields and initial configurations (here $5 \times 5$ system), (3) calculating spin relaxtion times

$(T_1, T_2$ and hetNOE) of the generated simulations, (4) ranking generated simulations based on quality evaluation against NMR spin relaxation data and selecting the best trajectories to compose the QEBSS ensemble, and (5) analyzing the selected QEBSS ensembles. The resulting ensembles are

considered to be plausible interpretations for conformational ensembles and dynamics resulting from the experimental spin relaxation times.

Most methods to determine ensembles of proteins with IDRs are based on generation of ensembles that reproduce experimental SAXS, chemical shift, paramagnetic relaxation enhancement (PRE), residual dipolar couplings (RDC), scalar couplings, and fluorescense resonance energy transfer (FRET) data using biased simulations or selecting suitable ensembles from a set of generated structures[54,55]. The main advantage of QEBSS to these is that it provides also dynamic information. Furthermore, spin relaxation times seem to be more sensitive to conformational ensembles than SAXS data (supplementary Fig. S16) and chemical shifts[17,55]. On the other hand, spin relaxation data is easier to measure, better available in the literature and can be connected to simulations more directly than PRE, FRET, or RDC data that require attachment of molecular probes or orientation of the sample[54].

Other methods to resolve dynamics of proteins with IDRs are based on intermediate models connecting MD to spin relaxation data[15] or reweighting segments of MD simulation trajectories to improve accuracy[56]. The main advantage of QEBSS to these is that it provides also conformational ensembles. Furthermore, MD simulations and NMR data are connected without any intermediate models in QEBSS. Reweighting segments of MD simulation trajectories to improve agreement with NMR spin relaxation times[56] is close to our approach. However, each fragment to be reweighted should be at least approximately 100 times longer than detected timescales[57]. This means that the minimum length of a fragment for the proteins studies in this work is approximately 1 μs for proteins studied in this work, setting stringent requirements for simulation lengths. Furthermore, the original trajectory should already be relatively realistic to enable finding correct result by reweighting. Advantage of QEBSS over reweighting a single trajectory is the more diverse set of simulation trajectories generated with different model parameters and initial configurations, which increases the probability to find ensembles in agreement with experiments and filling larger fraction of phase space.

We show that different force fields are required for the QEBSS approach because it is not a priori clear which force field would be most realistic for a given protein. As shown in this study, none of the used force fields outperforms others for the tested multidomain proteins. Ensembles resulting from QEBSS are composed of simulations with multiple force fields for all proteins except MANF for which all three selected trajectories are simulated with a99SB-disp parameters. However, also in this case, the two other a99SB-disp trajectories rank relatively low, indicating significant dependence on the initial structure.

Transferable force field providing accurate representations for all proteins and sufficiently long simulations, ensuring ergodic simulation trajectories would be highly desirable. Nevertheless, we demonstrate here that experimentally evaluated conformational ensembles and dynamics of multidomain proteins with IDRs can be achieved also with alternative strategy using currently available force fields and computational resources. In our conception, multiple force fields performing well for different types of systems in different conditions can replace the universal transferable force field, and a wide range of starting structures increase the sampled conformational space. QEBSS will then select the best solution among the produced data. However, it is important to note that QEBSS is constrained by the ability of available force fields to produce realistic options into the pool of ensembles that is used for selection. QEBSS is also not useful for systems with dynamic modes occurring on timescales slower than what can be reliably captured with good statistical accuracy in current MD simulations. Approximately 5 μs simulations of proteins with IDRs are currently feasible with state of the art resources, which provide good statistics for approximately 50 ns rotational timescales[57].

Our analysis reveals differences in conformational ensembles and dynamics between proteins that are inaccessible by other available methods paving the way for a deeper understanding of their biological function. Particularly, we observe significant differences in dynamic landscapes and conformational ensembles of two homologous proteins, CDNF and MANF, possibly also providing some explanation to the different biological

functions observed by these two proteins[50]. Interpretation of calmodulin conformational dynamics and ensembles from spin relaxation times with improved resolution to previous studies[12,13] can elucidate the important role of conformational changes in its biological functions[7]. The interpretative power of QEBSS is also demonstrated for the partially unfolded EN2 protein. However, previous simulations using mildly tuned parameters show slightly better agreement with experimental data[17], suggesting that improved force fields may be required for accurately modeling proteins with extended disordered regions.

QEBSS is also fully automatizable, enabling the generation of a large number of quality evaluated MD simulations that could be combined with machine learning algorithms to, for example, evaluate and improve force fields[38], or investigate how sequence correlates with protein biophysical properties. Related approaches are already being implemented for lipid bilayers[39].

## Conclusions

We present here the QEBSS protocol to find conformational ensembles with realistic dynamics for multidomain proteins. The approach is based on systematic generation of diverse set of conformational ensembles using MD simulations and subsequent selection of conformational ensembles with realistic dynamics using experimental spin relaxation data from NMR. We demonstrate the practical advance of QEBSS by determining conformational ensembles and dynamics of four multidomain (calmodulin, CDNF, MANF, and EN2) proteins with flexible disordered regions that are beyond the scope of other experimental methods to characterize protein dynamics.

Notably, the resulting ensemble from QEBSS is selected from simulations that are based on physical interactions between atoms, but its parameters or individual timescales have not been tuned to achieve good agreement with experiments. It can be therefore used to interpret how MD arises from atomistic resolution interactions and produces experimentally measured spin relaxation times. Dynamic landscapes and effective correlation times derived from QEBSS ensembles give comprehensible interpretations for protein dynamics. QEBSS concept can be automized and generally applied also to other types of flexible biomolecules than multidomain proteins studied here, such as IDPs, nanodiscs, and other protein-lipid complexes. Furthermore, we believe that QEBSS protocol will be highly useful in production of MD simulation data in good agreement with experiments, that is necessary for modern artificial intelligence based models predicting disordered protein properties[58,59].

## Methods
### QEBSS protocol

The idea of the QEBSS protocol is to select the most realistic conformational ensemble among a diverse set of MD trajectories generated with different force fields and initial configurations. The selection is based on evaluation of simulations against protein backbone $^{15}$N $T_1$ and $T_2$ spin relaxation times and hetNOE values from NMR experiments. QEBSS is implemented in five steps:

**1. Selection of starting structures.** The starting configurations of folded and two-domain proteins were obtained by selecting five different structures from experimental NMR ensembles. From each ensemble submitted, the first, last, and middle structures were selected for TonBCTD (PDB: 5LW8[22], models: 1, 5, 10, 15, 20), calmodulin (PDB: 1CFC[60], models: 1, 5, 10, 15, 20), CDNF (PDB: 4BIT[28], models: 1, 2, 4, 6, 8) and MANF (PDB: 2RQY[44], models: 1, 2, 4, 6, 8). The starting configurations for partially disordered EN2 were taken from 5 different time points (10 ps, 250 ns, 500 ns, 750 ns, 1000 ns) of a previously published MD trajectory starting from a fully extended linker[17,61].

**2. Running MD simulations.** From the five different starting configurations generated in step 1, simulations with five different force field parameters were started, resulting in a total of 25 simulations for each protein studied.

**Table 1 | Force fields, water models, Lennard-Jones cut-offs, and constraint algorithms**

| Force field | Lennard-Jones cutoff | Constraint algorithm[1] | Water model |
|---|---|---|---|
| a99SB-ILDN[40] | 1.0 | SHAKE | TIP4PEw[71] |
| DESamber[41] | 1.2 | SHAKE | TIP4PD[72] |
| a99SB-disp[21] | 1.2 | SHAKE | a99SBdisp[21] |
| aff03ws[42] | 1.4 | Lincs | TIP4P2005s[42] |
| a99SBws[42,73] | 1.4 | Lincs | TIP4P2005s[42,73] |

[1]Constraints were applied only on hydrogen bonds.

**Table 2 | Magnetic fields, ion concentrations, and temperatures**

| Protein | Magnetic field (MHz) | Ions | Temperature |
|---|---|---|---|
| TonBCTD[43] | 850 | 40 mM NaCl | 303 K |
| Calmodulin[60] | 600 | 100 mM KCl | 296 K |
| CDNF[28] | 600 | 100 mM NaCl | 298 K |
| MANF[44] | 600 | 20 mM NaCl | 310 K |
| EN2[45] | 800 | 40 mM NaCl | 303 K |

All systems were simulated using the following force field parameters: DESamber[41], amber99SB-disp (a99SB-disp)[21], amberff03ws (aff03ws)[42] and amber99SBws (a99SBws)[42], and amber99SB-ILDN (a99SB-ILDN)[40] with water models listed in Table 1. The first four force fields are optimized for disordered proteins, and the last previously reproduced spin relaxation times in good agreement with experiments for TonBCTD proteins[47]. As the first step, we checked that all force fields reproduce experimental spin relaxation times of folded TonBCTD. Notably, aff03ws and a99SBws force fields were originally parametrized using Langevin dynamics, but such simulations significantly overestimated $T_1$ spin relaxation times. Therefore, MD with a leap-frog algorithm is also used with these force fields. The used force fields, water models and simulation parameters are summarized in Table 1, These parameters resemble as closely as possible to the ones in original para-metrization articles and lead to a good agreement of spin relaxation times with experiments (Fig. 2) of TonBCTD.

Before starting the simulations, each protein was centered in a dode-cahedron (9 nm for CDNF) or cubic (6.5, 10, 12, and 22 nm for TonBCTD, calmodulin, MANF and EN2, respectively) simulation box using the Gro-macs function `gmx editconf`[62]. Solvent molecules were added to the system using Gromacs tool `gmx solvate` and systems were neutralized by adding sodium or chloride ions with `gmx genion`. Additional ions were added to all systems to mimic the ionic strength of the buffer used in the experimental NMR measurements (Table 2)[13,28,43–45]. Before production runs, system energies were minimized and short 4 ps pre-equilibration simulations were performed. Systems were simulated for 1 μs with periodic boundary conditions, 2 fs time step, velocity rescale temperature coupling, isotropic Parrinello-Rahman pressure coupling to 1 bar[63], and force fields specific settings defined in Table 1. A saving frequency of 10 ps was used. If clear conformational changes were observed in the radius of gyration within the first 300 ns of the simulation, this equilibration period was excluded from the analyses. All simulations were performed with Gromacs 2021.5 or 2020.1[64].

**3. Calculation of spin relaxation times and timescales.** Spin relaxation times, $T_1$ and $T_2$, and hetNOE of protein backbone $^{15}$N were calculated using Redfield equations as described previously[17,20,47]. Shortly, second order rotational correlation functions of N-H bonds were calculated from tra-jectories using `gmx rotacf` command in Gromacs[62]. Rotational correla-tion functions were then fitted to a sum of a large number, $N = 100$, of exponentially decaying functions with predefined timescales, $\tau_i$,

equidistantly spaced in logarithmic scale between 1 ps and 100 ns.

$$C_{\text{fit}}(t) = \sum_{i=1}^{N} \alpha_i e^{-t/\tau_i}, \quad (1)$$

where $\alpha_i$ is the weight with which the $i$-th exponential decay is present in the correlation function. Spectral densities, $J(\omega)$, were then calculated as an analytical Fourier transformation of Eq. (1) and substituted to Redfield equations using the same magnetic fields as in the corresponding experiments (Table 2). Redfield equations relate the spin relaxation rates to MD as[65,66]

$$\frac{1}{T_1} = \frac{d_{\text{NH}}^2 N_H}{20}[J(\omega_H - \omega_N) + 3J(\omega_N) \\ + 6J(\omega_H + \omega_N)] + \frac{(\Delta\sigma\omega_N)^2}{15}J(\omega_N) \quad (2)$$

$$\frac{1}{T_2} = \frac{1}{2}\frac{d_{\text{NH}}^2 N_H}{20}[4J(0) + J(\omega_H - \omega_N) + 3J(\omega_N) + 6J(\omega_H) \\ + 6J(\omega_H + \omega_N)] + \frac{(\Delta\sigma\omega_N)^2}{90}[4J(0) + 3J(\omega_N)] \quad (3)$$

$$\text{hetNOE} = 1 + \frac{d_{\text{NH}}^2 N_H}{20}[J(\omega_H - \omega_N) + 6J(\omega_H + \omega_N)]\frac{\gamma_H T_1}{\gamma_N}, \quad (4)$$

where $\omega_H$ and $\omega_N$ are Larmor frequencies of $^1$H and $^{15}$N, respectively, $N_H = 1$ is the number of protons in the N-H bond, and $\Delta\sigma = -160$ ppm is the chemical shift anisotropy[67]. The dipolar coupling constant is defined as $d_{NH} = \frac{\mu_0 \hbar \gamma_H \gamma_N}{4\pi \langle r_{NH}^3 \rangle}$, where $\mu_0$ is vacuum permeability, $\hbar$ is the reduced Planck constant, $\gamma_H$ and $\gamma_N$ are the gyromagnetic constants of $^1$H and $^{15}$N, respectively, and the average cubic length is calculated as $\langle r_{NH}^3 \rangle = (0.101 \text{ nm})^3$. Spectral density, $J(\omega)$, is the Fourier transform of the second-order rotational correlation function of N-H bond vector,

$$g(t) = \left\langle \frac{3}{2}\cos^2\theta_{t'+t} - \frac{1}{2} \right\rangle_{t'}, \quad (5)$$

where $\theta_{t'+t}$ is the angle between N-H bond vector at the times $t$ and $t'$. The spin relaxation time calculation is implemented in the python code available at https://github.com/nencini/NMR_FF_tools/tree/master/relaxation_times. The weights for timescales resulting from the fitting of Eq. (1) were used to interpret the dynamic landscape of proteins[17,20]. Effective correlation times $\tau_{\text{eff}}$ were calculated as an integral over the correlation function fit

$$\tau_{\text{eff}} = \sum_{i=1}^{N} \alpha_i \tau_i. \quad (6)$$

**4. Ranking of simulations and selection of best ensembles.** The qua-lities of different spin relaxation times in each simulation were first evaluated by calculating RMSDs from experimental spin relaxation times separately for each spin relaxation time, $T_1$, $T_2$, and hetNOE, by aver-aging over residues. Comparison numbers for each spin relaxation time were then calculated for each simulation by dividing the RMSD of a simulation in question with the smallest RMSD for that spin relaxation time observed in all simulations for the given protein. Such evaluation based on spin relaxation times is sensitive for both conformational ensemble and dynamic landscape of partially disordered proteins[17].

We then selected simulations that have comparison numbers for all spin relaxation times below 150% (i.e., deviating less than 50% from the best simulation) for further analysis. Considering only simulations with good agreement simultaneously for all $T_1$, $T_2$, and hetNOE values should lead to an interpretation that is valid for a wide range of rotational motions because different spin relaxation times are sensitive to different timescales.

In principle, the used criteria with simulations performed in this work may lead to the selection of 0–25 trajectories into the QEBSS ensemble. Because good agreement simultaneously for all spin relaxation times is required, a poor comparison number for one or two spin relaxation times leads to the rejection. Therefore, it is possible that none of the trajectories would be selected and simulations with the best comparison numbers for individual spin relaxation times may not be among the selected simulations. In practice, we set the criteria such that at least one simulation satisfied them for all systems studied in this work. Notably, the selected trajectories may originate from different force fields. Nevertheless, incompatibility of simulation parameters between trajectories within QEBSS ensemble is not an issue because they are not simulated further.

**5. Characterization of the selected QEBSS ensemble.** After selecting the simulations that best represent the experimental data, interpretation is provided by analysing average properties over the selected simulation trajectories. Average properties of ensembles (such as radius of gyration and backbone orientational correlation) are calculated by averaging all the frames in selected trajectories. For analyses including rotational correlation functions (such as spin relaxation times, dynamical timescales and effective correlation times), the average correlation functions over selected simulations are first calculated for each residue, and these correlation functions are then used in the analyses.

Radii of gyrations were calculated using `gmx gyrate` from Gromacs[62]. Overlayed snapshot figures were made with PyMOL software[68] by selecting 10 frames with equal spacing from the trajectories, and superposing the N-terminal folded domain of calmodulin, CDNF and MANF, C-terminal folded domain of EN2, and the whole TonBCTD protein. Orientational correlation between different residues in protein backbone was analyzed by calculating the dot products of vectors connecting the $C^\alpha$ carbons in neighboring residues, $\langle v_i \cdot v_j \rangle$, where $v_i$ is normalized vector between $C_i^\alpha$ and $C_{i+1}^\alpha$ [17]. Positive or negative values of the averaged and normalized dot product indicate a positive or negative orientational correlation between the residues, respectively, while completely random orientations give zero. The average minimum distance between each residue, selecting all atoms, was calculated with 1 ns intervals using the Gromacs function `mindist`[62]. The PCA was performed using the GROMACS tools `gmx covar` and `gmx anaeig`[62]. For each protein, all 25 simulation trajectories were first concatenated using `gmx trjcat` to create a combined trajectory, which was used as the input for `gmx covar` to calculate the covariance matrix of $C_\alpha$ atomic positional fluctuations. This ensured that the principal components (eigenvectors) were derived from the full conformational space sampled across all replicas. Subsequently, individual trajectories were projected onto the first and second eigenvectors using `gmx anaeig`.

**Calculation of SAXS intensities from MD simulations**
Every 1000th frame of the trajectories were extracted as a pdb file using the `gmx trjconv` from Gromacs[62]. SAXS intensities were calculated for each pdb structure separately using crysol (version 3.2.0)[69] with the options: –smax 0.5 –ns 501 –lm 20 –dns 0.334 –dro 0.03 –explicit-hydrogens –shell directional –fb 17. SAXS intensities for each simulation were then calculated by averaging over intensities from individual structures. As magnitudes of experimental intensity values are arbitrary, SAXS profiles from MD simulations were fitted to the experimental data using the Levenberg-Marquardt algorithm as ($q$ values below 0.01 and above 0.3 were filtered out)

$$I_{fit} = scale \times I_{comp} + offset. \tag{7}$$

Both experimental and fitted intensity data were normalized for comparative analysis by subtracting the minimum intensity and scaling by the intensity range:

$$I_{normalized} = \frac{I - I_{min}}{I_{max} - I_{min}} \tag{8}$$

Quality of SAXS intenstity from simulations against experiments was then evaluated using RMSE values

$$RMSE = \sqrt{\frac{1}{N} \sum_{i=1}^{N} \left( I_{exp,i} - I_{fit,i} \right)^2}. \tag{9}$$

**SAXS experiments**
The buffer of recombinant human MANF protein (P-101-100, Icosagen) was exchanged by dialysis to 10 mM sodium phosphate buffer with 2% glycerol at pH 6.0 with a final concentration of 0.85 mg/ml. The recombinant human CDNF (P-100-100, Icosagen) was dialysed to phosphate buffered saline with 2% glycerol at pH 7.4 and concentration of 3 mg/ml. SAXS data were collected at ESRF beam line BM29 at wavelength of 0.99186 Å with total sample volume of 50 μl for each protein at 20 °C, with 1 s exposure time per image with ten repeats per sample, and data were averaged and buffer subtracted for further analysis.

## Data availability
All simulation data is available in Zenodo repositories listed in Supplementary Table S6. Numerical data used in figures is available at ref. 70.

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

## Acknowledgements

We acknowledge CSC—IT Center for Science for computational resources and the Research Council of Finland for funding (grant nos. 315596, 319902, 345631, 356568, and 350636). RN acknowledges funding from Emil Aaltonen Foundation and E.Y. from EC Research Innovation Action HPC-EUROPA3 (INFRAIA-2016-1-730897) under the H2020 Programme. T.K. acknowledges funding from Jane and Aatos Erkko Foundation. We thank Prof. Mikko Airavaara for providing MANF and Prof. Mart Saarma for providing the CDNF protein for SAXS experiments. ESRF BM29 SAXS beam line staff is gratefully acknowledged for help with measurement of the samples.

## Author contributions

A.E.S. Performed and analyzed simulations, wrote the manuscript together with O.H.S.O., and contributed in conceptualization of the work. R.N. supported in implementation of simulation analyses, particularly for spin relaxation times, and contributed to editing the manuscript. E.Y. performed preliminary calmodulin simulations that supported the conceptualization of the work. T.K. and S.F. provided and analysed SAXS data, and contributed to editing the manuscript. V.S. provided samples for SAXS experiments, and contributed to editing the manuscript. O.H.S.O. conceptualized and supervised the work, and wrote the manuscript together with A.E.S.

## Competing interests

The authors declare no competing interests.
