## [Transparent Peer Review file · Communications Chemistry]

Quality Evaluation Based Simulation Selection (QEBSS) for analysis of conformational ensembles and dynamics of multi-domain proteins

Corresponding Author: Dr Samuli Ollila

Version 0:

Reviewer comments:

Reviewer #1

(Remarks to the Author)

In this paper, the authors proposed QEBSS to analyze conformational ensembles and dynamics of multidomain proteins integrating multiple MD trajectories based on different force fields with NMR experimental data (T1, T2 spin relaxation, and hetNOE). In their protocol, as far as I understood, they first performed MD simulations with different force fields, ranked simulation trajectories compared to NMR data, and selected only the trajectories to construct the QEBSS ensemble.

This work is interesting, but I have many technical questions on constructing the QEBSS ensemble and some data they showed.

- 1) Did they mix MD simulation trajectories with different force fields to construct the QEBSS ensemble? Table S3-7 shows all the data about matching with NMR relaxation parameters. The number of QEBSS trajectories differs for the five proteins. How did they determine it?
- 2) Besides the fitting with the NMR relaxation parameters, how similar are the selected MD trajectories? In Figures S6-10, they showed protein backbone orientation maps. Those in the selected trajectories differ significantly, while other trajectories look similar.
- 3) I first read the main text but could not understand the details of the approach and meaning of the figures (Figs. 4 and 5). After reading SI, I found that sufficient information is given. I suggest the authors write essential technical information compactly in the main text. Otherwise, it isn't easy to understand the details of this work.
- 4) Though the QEBSS ensemble can reproduce NMR relaxation parameters, mixing MD simulations with different force fields should be done most carefully. One reason is the use of varying water models in different force fields, which suggests that protein-water interactions are not the same in these simulations. Cutoff and Constraint algorithms also differ between the five force fields. From a computational point of view, I hesitated to mix the trajectories with different force fields for interpreting structural ensembles of proteins. However, the authors' group seems to focus on the ensemble generations that match the NMR relaxation parameters. I would like to hear the authors' opinion about the potential uses of the QEBSS ensembles and the limitations in the QEBSS.

Reviewer #2

(Remarks to the Author)

Dear Editor,

First of all, I would like to thank you for giving me the opportunity to review the manuscript "QEBSS: Quality Evaluation Based Simulation Selection for analysis of conformational ensembles and dynamics of multidomain proteins".

The authors introduce the QEBSS protocol (Quality Evaluation Based Simulation Selection) to interpret conformational ensembles and dynamics of multi-domain proteins using a combination of MD simulations and protein backbone ¹⁵N T1 and T2 spin relaxation times and hetNOE values from NMR experiments. Despite its conceptual simplicity, this approach could be very useful in the field of conformational ensembles.

The manuscript is well written, constructed and illustrated.

Some minor points:

- The results exposed in this manuscript depends substantially on the MD simulations performed at the beginning of the approach, in section 2.1. For the sake of reproducibility, this section should be more detailed, and provide a full description of the setup of the MD simulations, and not just the force field: how were chosen the 5 starting structures? What integrator was used? What was the time step? How the effect of water was included? Were some constraints applied on some bonds? Temperature of the simulation? Etc.
- How were the structural superimpositions done in Fig 1 and Fig 4?

Reviewer #3

(Remarks to the Author)

The authors describe a molecular dynamics simulation methodology to generate ensembles or multidomain proteins containing intrinsically disordered regions that best match a set of NMR measured data. Although in principle interesting, I have several concern about the current version that need to be carefully addressed before I can recommend the work for publication.

First, the filtering threshold of 50% RMSD agreement seems rather generous. Indeed. large remaining discrepancies between QEBSS ensembles and experiment in Fig 3, questioning the limitations of the proposed ensembles.

In that regard, cross validation of the ensembles with independent data like RDCs would be highly desirable. Additionally, it should be assessed how the presented ensembles compare to reference structures/ensembles of the studied proteins that are present in the PDB.

In addition, I was missing an analysis of the output ensembles (compared to the input ensembles), for example based on principal components analysis to assess what is the effect of filtering. In particular, it would allow to address the important issue if multiple "best" ensembles reproducibly converge to the same stable answer.

The manuscript mentions that "Reweighting segments of MD simulation trajectories is close to our approach", but the appropriate reweighting references are not cited.

Finally, "dynamic landscapes" are mentioned in the manuscript but not exploited: the manuscript implies that multiple trajectories show a closer match the experimental data (outperforming the best single basin), but which (populations of which) conformations are the basis of that?

minor:

- "ensembles that predict experimental spin relaxation times"  that reproduce

- "overperforms"  outperforms

Version 1:

Reviewer comments:

Reviewer #1

(Remarks to the Author)

The revised manuscript has been improved sufficiently. The authors' replies to the reviewers' comments are fine.

Reviewer #2

(Remarks to the Author)

Dear Editor,

The authors have addressed my questions satisfactorily, and I have no further comments or concerns.

Sincerely,

Reviewer #3

(Remarks to the Author)

The authors have satisfactorily addressed my concerns.

Reviewer #1:

In this paper, the authors proposed QEBSS to analyze conformational ensembles and dynamics of multidomain proteins integrating multiple MD trajectories based on different force fields with NMR experimental data (T1, T2 spin relaxation, and hetNOE). In their protocol, as far as I understood, they first performed MD simulations with different force fields, ranked simulation trajectories compared to NMR data, and selected only the trajectories to construct the QEBSS ensemble.

This work is interesting, but I have many technical questions on constructing the QEBSS ensemble and some data they showed.

1) Did they mix MD simulation trajectories with different force fields to construct the QEBSS ensemble? Table S3-7 shows all the data about matching with NMR relaxation parameters. The number of QEBSS trajectories differs for the five proteins. How did they determine it?

AUTHOR REPLY:

We thank the reviewer for positive and constructive feedback.

The QEBSS protocol is designed to find most realistic ensembles among diverse MD simulations data created with different starting conformations and force fields. To construct the QEBSS ensemble, we first evaluated how well each trajectory matches experimental NMR spin relaxation data (T1, T2, and hetNOE) by calculating the root mean square deviation (RMSD) between simulated and experimental values.

For each NMR parameter, the trajectory with the lowest RMSD is used as a reference and assigned a 100% match score. Trajectories are then included in the QEBSS ensemble if their RMSD values are within 50% of the best trajectory simultaneously for **all three** parameters (T1, T2, and hetNOE). This fixed threshold ensures consistent selection criteria across all proteins. I may also result in different numbers of trajectories selected for each case and combinations of trajectories originating from different force fields.

To clarify these and other aspects of QEBSS protocol, we have revised the Methods section by moving it to the main text and restructuring it into numbered steps. Particularly, the number of trajectories and mixtures of different force fields are now clarified in Methods section:

“In principle, the used criteria with simulations performed in this work may lead to the selection of 0-25 trajectories into the QEBSS ensemble. Because good agreement simultaneously for all spin relaxation times is required, a poor comparison number for one or two spin relaxation times leads to the rejection. Therefore, it is possible that none of the trajectories would be selected and simulations with the best comparison numbers for individual spin relaxation times may not be among the selected simulations. In

practice, we set the criteria such that at least one simulation satisfied them for all systems studied in this work. Notably, the selected trajectories may originate from different force fields. Nevertheless, incompatibility of simulation parameters between trajectories within QEBSS ensemble is not an issue because they are not simulated further.”

To further clarify the manuscript, we have also separated the Results and Discussion to separate sections, and divided section 2.3. into subsections.

2) Besides the fitting with the NMR relaxation parameters, how similar are the selected MD trajectories? In Figures S6-10, they showed protein backbone orientation maps. Those in the selected trajectories differ significantly, while other trajectories look similar.

AUTHOR REPLY:

Characterization and mutual comparison of ensembles of flexible proteins is not straightforward. In the manuscript, we have analyzed radius of gyration distributions, backbone orientation maps, and minimum distance maps from all 25 trajectories generated for each protein. While some of these measures exhibit similarities between selected trajectories, others may not.

To better quantify the similarities and differences between the trajectories, we performed a principal component analysis (PCA) for all trajectories and proteins (as suggested by another reviewer). Results from PCA analysis are shown below and in Supplementary Figure S18. The results are now discussed in the second last paragraph of section 2.2:

“Because characterization and mutual comparisons of conformational ensembles is complicated, differences between ensembles accepted or rejected by QEBSS are not always straightforward to explain. Nevertheless, distinctive features in radius of gyration distributions are observed in all proteins except CDNF, where multiple types of distributions appear across the selected simulations (Supplementary Figs.S1–S5). To quantify differences between simulation ensembles, we performed principal component analysis (PCA) for all simulations (Supplementary Fig. S18). For EN2 and calmodulin, their selected ensembles are closely positioned in PCA space, indicating structural similarity (Supplementary Fig. S18). Substantial similarities for the second PC component of QEBSS ensembles of MANF are observed, while CDNF ensembles are more diverse as observed also for radius of gyration distributions. As expected, variance

in PCA analysis for folded TonBCTD is substantially smaller than for proteins with flexible IDRs."

Figure S18: Principal component analysis of all 25 MD simulations and the QEBSS ensemble for TonBCTD, calmodulin, CDNF, MANF and EN2. Simulations selected for the QEBSS ensemble are highlighted with a red circle.

3) I first read the main text but could not understand the details of the approach and meaning of the figures (Figs. 4 and 5). After reading SI, I found that sufficient information is given. I suggest the authors write essential technical information compactly in the main text. Otherwise, it isn't easy to understand the details of this work.

AUTHOR REPLY:

We have now moved the Methods section into the main text and restructured it into numbered steps. We have also clarified methodological details in the main text and refer to the numbered steps of QEBSS protocol described in Methods. We believe these revisions substantially improve the understandability of our manuscript.

4) Though the QEBSS ensemble can reproduce NMR relaxation parameters, mixing MD simulations with different force fields should be done most carefully. One reason is the use of varying water models in different force fields, which suggests that protein-water interactions are not the same in these simulations. Cutoff and Constraint algorithms also differ between the five force fields. From a computational point of view, I hesitated to mix the trajectories with different force fields for interpreting structural ensembles of proteins. However, the authors' group seems to focus on the ensemble generations that match the NMR relaxation parameters. I would like to hear the authors' opinion about the potential uses of the QEBSS ensembles and the limitations in the QEBSS.

AUTHOR REPLY:

We fully agree that compatibility of used cutoffs, constraint algorithms and water models with the used force fields must be carefully considered when performing MD simulations. If this is not done, semi-empirical forces between atoms will be inconsistently calculated during the MD simulation steps, which may lead to spurious results. Therefore, we have set these parameters to closely resemble the ones used in original parameterizations of used force fields when running the simulations to produce the pool of ensembles from which the best ensembles are selected by the QEBSS (step 2 in QEBSS protocol described in the methods).

After running the simulations, QEBSS protocol selects the most realistic trajectories based on comparison with NMR data and combines them to the QEBSS ensemble (step 4 in QEBSS protocol described in the methods of the revised manuscript). The selection criteria for the QEBSS ensemble are based on quality of protein backbone dynamics with respect to experiments. QEBSS ensembles can contain data from different force fields with different parameters, but these trajectories are not simulated further. Therefore, forces between atoms are not calculated after the QEBSS selection and compatibility of parameters within simulations used to compose QEBSS ensemble is not important. These points are now clarified in the description of step 4 in QEBSS protocol in Methods section:

“Notably, the selected trajectories may originate from different force fields. Nevertheless, incompatibility of simulation parameters between trajectories within QEBSS ensemble is not an issue because they are not simulated further.”

Nonetheless, the main limitations of QEBSS arise from accuracy of currently available force fields and achievable timescales in MD simulations. These are now discussed in the Discussion section:

“However, it is important to note that QEBSS is constrained by the ability of available force fields to produce realistic options into the pool of ensembles that is used for selection. QEBSS is also not useful for systems with dynamic modes occurring on timescales slower than what can be reliably captured with good statistical accuracy in current MD simulations. Approximately 5 μ s simulations of proteins with IRDs are currently feasible with state of the art resources, which provide good statistics for approximately 50 ns rotational timescales [58].”

Reviewer #2:

The authors introduce the QEBSS protocol (Quality Evaluation Based Simulation Selection) to interpret conformational ensembles and dynamics of multi-domain proteins using a combination of MD simulations and protein backbone ^{15}N T1 and T2 spin relaxation times and hetNOE values from NMR experiments. Despite its conceptual simplicity, this approach could be very useful in the field of conformational ensembles.

The manuscript is well written, constructed and illustrated.

AUTHOR REPLY:

We thank the reviewer for the positive feedback and for recognizing the usefulness of our approach for conformational ensemble generation.

Some minor points:

The results exposed in this manuscript depends substantially on the MD simulations performed at the beginning of the approach, in section 2.1. For the sake of reproducibility, this section should be more detailed, and provide a full description of the setup of the MD simulations, and not just the force field: how were chosen the 5 starting structures? What integrator was used? What was the time step? How the effect of water was included? Were some constraints applied on some bonds? Temperature of the simulation? Etc.

AUTHOR REPLY:

We agree that the details mentioned are highly relevant. However, we are afraid that including all the details in section 2.1. would make it cumbersome. To clarify the methodological details, we have now moved the Methods section into the main text and restructured it into numbered steps. We have also clarified methodological details in the main text and refer to the numbered steps of QEBSS protocol described in Methods. We have also ensured that all the mentioned relevant details are given in the Methods section. We believe these revisions substantially improve the understandability of methodological details in our manuscript.

To further clarify the manuscript, we have also separated the Results and Discussion to separate sections, and divided section 2.3. into subsections.

- How were the structural superimpositions done in Fig 1 and Fig 4?

AUTHOR REPLY:

We have further clarified the generation of the structural superimpositions in the legends of Figure 1 and Figure 4:

“Fig. 1 Average radius of gyration (nm) for each force field calculated from five replicas (left). Representative snapshots showing 50 overlaid structures per force field, with 10 equally spaced structures taken from each of the five replicas. For structural alignment, the entire protein was used for TonBCTD, while the N-terminal folded domains were used for calmodulin, CDNF and MANF, and the C-terminal domain for EN2. Domain organization, sequence, and residue numbering of each protein are shown to illustrate the different domains of the multidomain proteins (right).”

“Fig. 4 Radius of gyration (R_g in nm) distributions (left) snapshots displaying 10 overlaid structures (middle) and timescales (right) from QEBSS ensembles for different proteins. Dashed vertical lines show the average R_g and the experimental value for calmodulin is

marked with a black dashed line. For the snapshots structural alignment of the entire protein was used for TonBCTD, while the N-terminal folded domains were used for calmodulin, CDNF, and MANF, and the C-terminal domain for EN2. The point sizes represent the weight of each timescale in the rotational relaxation process.”

Further details are available in the Methods section:

“Overlaid snapshot figures were made with PyMOL software by selecting 10 frames with equal spacing from the trajectories, and superposing the N-terminal folded domain of calmodulin, CDNF and MANF, C-terminal folded domain of EN2, and the whole TonBCTD protein.”

Reviewer #3:

The authors describe a molecular dynamics simulation methodology to generate ensembles or multidomain proteins containing intrinsically disordered regions that best match a set of NMR measured data. Although in principle interesting, I have several concern about the current version that need to be carefully addressed before I can recommend the work for publication.

AUTHOR REPLY:

We thank the reviewer for the thoughtful comments and constructive feedback. We believe that we have been able to address all the concerns and the manuscript is now substantially improved.

First, the filtering threshold of 50% RMSD agreement seems rather generous. Indeed, large remaining discrepancies between QEBSS ensembles and experiment in Fig 3, questioning the limitations of the proposed ensembles.

AUTHOR REPLY:

We chose the 50% RMSD threshold to ensure that at least one simulation would be selected for QEBSS ensemble for each protein for analysis. We agree that there are discrepancies between spin relaxation times from QEBSS ensemble and experiments for EN2 and MANF residues 100-171 in Fig. 3. Nevertheless, general agreement with experiments is still very good. Notably, the RMSDs are generally lower for the QEBSS ensembles than single trajectory simulations.

In conclusion, the limitations of QEBSS are mainly related to the accuracy of available force fields and achievable simulation timescales rather than the exact selection criteria. If more accurate force fields become available in the future, the selection criteria can be then tightened to result in better QEBSS solutions.

We have now clarified these aspects in section 2.2:

"... In this study, we set the QEBSS threshold at 50 % to ensure that each protein system had at least one simulation included in the QEBSS ensemble. Future studies may benefit from optimizing this threshold based on the quality of simulation and experimental agreement.

Overall agreement of QEBSS ensembles and experiments is very good in Fig. 3 with the exceptions of EN2 simulations where T2 values are slightly underestimated and hetNOE values overestimates for the flexible linker, and of MANF for which some deviations are observed for residues 100-171. These differences highlight the importance of developing more accurate force field parameters that would create more realistic ensembles for the QEBSS protocol. ..."

To further clarify the manuscript, we have also separated the Results and Discussion to separate sections, and divided section 2.3. into subsections.

In that regard, cross validation of the ensembles with independent data like RDCs would be highly desirable.

AUTHOR REPLY:

We have cross validated QEBSS ensembles against SAXS data and spin relaxation times measured with magnetic field strengths that were not used in the selection of the ensemble (supplementary figures S16-S17). Additional cross validation against RDC data would be definitely interesting and potentially useful. However, we are not aware of available experimental RDCs values for proteins studied here measured at comparable experimental conditions. RDC values for calcium-bound calmodulin are available (<https://www.nature.com/articles/nsb1101-990>) but our QEBSS ensemble is resolved for calcium free calmodulin. Also, the published RDC values for calcium bound calmodulin are not assigned to specific residues.

Furthermore, RDCs are typically measured from samples where proteins have been ordered with the help of larger molecules in media. When comparing such RDC values with MD simulations, the ordering due to media has to be somehow taken into account (<https://doi.org/10.1016/j.jmr.2008.01.007>). Due to these complications, comparison of RDC values between simulations and experiments is less direct than for spin relaxation times or SAXS.

Additionally, it should be assessed how the presented ensembles compare to reference structures/ensembles of the studied proteins that are present in the PDB.

AUTHOR REPLY:

We thank the reviewer for the relevant suggestion. We have now included analyses of radius of gyration distributions, backbone orientational correlation maps and minimum distance maps from the reference structures available at Protein Data Bank (PDB) (supplementary figure S19). For each of the studied proteins—TonBCTD (5LW8),

calmodulin (1CFC), CDNF (4BIT), and MANF (2RQY)—we identified 10-20 reference PDB structures. The results are now commented in the end of section 2.2:

“Furthermore, the selected QEBSS ensembles differ significantly from the ensemble used to select the starting structures (Supplementary Fig. S19) supporting the conclusion that the QEBSS protocol provides new insights into protein ensembles and improves the interpretation of experimental spin relaxation data.”

Regarding the EN2 protein, the starting structures were derived from an existing molecular dynamics trajectory, so we did not include EN2 in the comparison with reference PDB ensembles.

Figure S19: Radius of gyration (R_g in nm) distributions (left), protein backbone orientation

correlation maps for vectors between C α carbons of consecutive residues (middle), average minimum distance between residues maps (right) of the starting structure ensembles from RCSB for TonBCTD (PDB: 5LW87), calmodulin (PDB: 1CFC8), CDNF (PDB: 4BIT3) and MANF (PDB: 2RQY4).

In addition, I was missing an analysis of the output ensembles (compared to the input ensembles), for example based on principal components analysis to assess what is the effect of filtering. In particular, it would allow to address the important issue if multiple "best" ensembles reproducibly converge to the same stable answer.

AUTHOR REPLY:

We thank the reviewer for this excellent suggestion. We have now performed a principal component analysis (PCA) of all 25 simulated trajectories for each protein, as well as on the QEBSS ensembles. This analysis has been included as Supplementary Figure S18, and we have also copied the figure below along with the corresponding discussion from section 2.2:

“To quantify differences between simulation ensembles, we performed principal component analysis (PCA) for all simulations (Supplementary Fig. S18). For EN2 and calmodulin, their selected ensembles are closely positioned in PCA space, indicating structural similarity (Supplementary Fig. S18). Substantial similarities for the second PC component of QEBSS ensembles of MANF are observed, while CDNF ensembles are more diverse as observed also for radius of gyration distributions. As expected, variance in PCA analysis for folded TonBCTD is substantially smaller than for proteins with flexible IDRs.”

The manuscript mentions that "Reweighting segments of MD simulation trajectories is close to our approach", but the appropriate reweighting references are not cited.

AUTHOR REPLY:

We thank the reviewer for pointing this out. We have now added the correct citation and clarified this paragraph. It now reads as:

“Reweighting segments of MD simulation trajectories to improve agreement with NMR spin relaxation times [57] is close to our approach. However, each fragment to be reweighted should be at least approximately 100 times longer than detected timescales [58]. This means that the minimum length of a fragment is approximately 1 μ s for proteins studied in this work, setting stringent requirements for simulation lengths. Furthermore, the original trajectory should already be relatively realistic to enable finding correct result by reweighting. Advantage of QEBSS over reweighting a single trajectory is the more diverse set of simulation trajectories generated with different

model parameters and initial configurations, which increases the probability to find ensembles in agreement with experiments and filling larger fraction of phase space."

Reference [57] is article presenting the reweighting approach and [58] demonstrates that a trajectory should be 100 times longer than the detected timescale

[57] N. Salvi, A. Abyzov, M. Blackledge, J. Phys. Chem. Lett. 2016, 7, 2483.

[58] C.-Y. Lu, D. A. Vanden Bout, J. Chem. Phys. 2006, 125, 124701.

Finally, "dynamic landscapes" are mentioned in the manuscript but not exploited: the manuscript implies that multiple trajectories show a closer match the experimental data (outperforming the best single basin), but which (populations of which) conformations are the basis of that?

AUTHOR REPLY:

We exploit the dynamic landscapes in section 2.3 to characterize differences in conformational ensembles and dynamics between different proteins.

We agree with the reviewer that dynamic landscapes could be also used to clarify which conformational populations lead to dynamics that improve agreement of QEBSS ensemble with experiments. However, such analyses are relatively complicated and have not yet been reported in the literature. Incorporating them would significantly lengthen an already extensive manuscript. Therefore, we believe that it is better to leave these studies for the future works.

minor:

**-"ensembles that predict experimental spin relaxation times"  that reproduce
-"overperforms"  outperforms**

AUTHOR REPLY:

These have been corrected in the manuscript.

REVIEWERS' COMMENTS:

Reviewer #1 (Remarks to the Author):

The revised manuscript has been improved sufficiently. The authors' replies to the reviewers' comments are fine.

Reviewer #2 (Remarks to the Author):

Dear Editor,

The authors have addressed my questions satisfactorily, and I have no further comments or concerns.

Sincerely,

Reviewer #3 (Remarks to the Author):

The authors have satisfactorily addressed my concerns.

AUTHOR REPLY:

We thank reviewers' for the careful evaluation of the manuscript and are happy to hear that they are satisfied with our answers.